# Different forms of superspreading lead to different outcomes: Heterogeneity in infectiousness and contact behavior relevant for the case of SARS-CoV-2

Elise J. Kuylen[1,2]*, Andrea Torneri[1], Lander Willem[1], Pieter J. K. Libin[2,3,4], Steven Abrams[2,5], Pietro Coletti[2], Nicolas Franco[2,6], Frederik Verelst[1¤], Philippe Beutels[1,7], Jori Liesenborgs[8], Niel Hens[1,2]

**1** Centre for Health Economic Research and Modeling Infectious Diseases, University of Antwerp, Antwerp, Belgium, **2** Data Science Institute, I-BioStat, Hasselt University, Hasselt, Belgium, **3** Artificial Intelligence Lab, Vrije Universiteit Brussel, Brussels, Belgium, **4** Rega Institute for Medical Research, Clinical and Epidemiological Virology, University of Leuven, Leuven, Belgium, **5** Global Health Institute, University of Antwerp, Antwerp, Belgium, **6** Namur Institute for Complex Systems, Department of Mathematics, University of Namur, Namur, Belgium, **7** School of Public Health and Community Medicine, The University of New South Wales, Sydney, NSW, Australia, **8** Expertise Centre for Digital Media, Hasselt University - transnational University Limburg, Hasselt, Belgium

¤ Current address: GSK, Wavre, Belgium
* elise.kuylen@uantwerpen.be

**Data Availability Statement:** The code of STRIDE is open-source and available at https://github.com/elisekaa/stride. All code and data used for the

## Abstract

Superspreading events play an important role in the spread of several pathogens, such as SARS-CoV-2. While the basic reproduction number of the original Wuhan SARS-CoV-2 is estimated to be about 3 for Belgium, there is substantial inter-individual variation in the number of secondary cases each infected individual causes—with most infectious individuals generating no or only a few secondary cases, while about 20% of infectious individuals is responsible for 80% of new infections. Multiple factors contribute to the occurrence of superspreading events: heterogeneity in infectiousness, individual variations in susceptibility, differences in contact behavior, and the environment in which transmission takes place. While superspreading has been included in several infectious disease transmission models, research into the effects of different forms of superspreading on the spread of pathogens remains limited. To disentangle the effects of infectiousness-related heterogeneity on the one hand and contact-related heterogeneity on the other, we implemented both forms of superspreading in an individual-based model describing the transmission and spread of SARS-CoV-2 in a synthetic Belgian population. We considered its impact on viral spread as well as on epidemic resurgence after a period of social distancing. We found that the effects of superspreading driven by heterogeneity in infectiousness are different from the effects of superspreading driven by heterogeneity in contact behavior. On the one hand, a higher level of infectiousness-related heterogeneity results in a lower risk of an outbreak persisting following the introduction of one infected individual into the population. Outbreaks that did persist led to fewer total cases and were slower, with a lower peak which occurred at a later point in time, and a lower herd immunity threshold. Finally, the risk of resurgence of an

research in this paper has been assigned a DOI using Zenodo: 10.5281/zenodo.6669350.

**Funding:** We acknowledge funding from the Flemish Government through IBOF (E.J.K., P.B. and N.H.: IBOF.21.027 Descartes project) and from the Research Foundation Flanders (FWO) (S.A. and N.H.: G0G2920N RESTORE project; E.J.K and N.H.: grant agreement R-7861). This work also received funding from the European Research Council (ERC) under the European Union's Horizon 2020 research and innovation program (A.T., P.J.K.L., N. F., F.V., P.B and N.H.: grant agreement 101003688 EpiPose; P.C. and N.H.: grant agreement 682540 TransMID). Furthermore, L.W. and P.J.K.L. gratefully acknowledge support from the Research Foundation Flanders (FWO) via postdoctoral fellowships 1234620N (L.W.) and 1242021N (P.J. K.L.). P.J.K.L. also acknowledges support from the Research council of the Vrije Universiteit Brussel (OZR-VUB) via grant number OZR3863BOF, and from the Flemish Government through the AI Research Program. Finally, P.B. and N.H. acknowledge funding from University of Antwerp via the Antwerp Study Center for Infectious Diseases (ASCID) and the Methusalem-Centre of Excellence consortium VAX–IDEA. The funders had no role in study design, data collection and analysis, decision to publish, or preparation of the manuscript.

**Competing interests:** I have read the journal's policy and the authors of this manuscript have the following competing interests: FV contributed to this work as a full time employee of the University of Antwerp. As of 21 March 2022, after his contributions to this work ended, FV is employed by the GSK group of companies.

outbreak following a period of lockdown decreased. On the other hand, when contact-related heterogeneity was high, this also led to fewer cases in total during persistent outbreaks, but caused outbreaks to be more explosive in regard to other aspects (such as higher peaks which occurred earlier, and a higher herd immunity threshold). Finally, the risk of resurgence of an outbreak following a period of lockdown increased. We found that these effects were conserved when testing combinations of infectiousness-related and contact-related heterogeneity.

## Author summary

To investigate the effect of different sources of superspreading on disease dynamics, we implemented superspreading driven by heterogeneity in infectiousness and heterogeneity in contact behavior into an individual-based model for the transmission of SARS-CoV-2 in the Belgian population. We compared the impact of both forms of superspreading in a scenario without interventions as well as in a scenario in which a period of strict social distancing (i.e. a lockdown) is followed by a period of partial release. We found that both forms of superspreading have very different effects. On the one hand, increasing the level of infectiousness-related heterogeneity led to less outbreaks being observed following the introduction of one infected individual in the population. Furthermore, final outbreak sizes decreased, and outbreaks became slower, with lower and later peaks, and a lower herd immunity threshold. Finally, the risk for resurgence of an outbreak following a period of lockdown also decreased. On the other hand, when contact-related heterogeneity was high, this also led to smaller final sizes, but caused outbreaks to be more explosive regarding other aspects (such as higher peaks that occurred earlier). The herd immunity threshold also increased, as did the risk of resurgence of outbreaks.

## Introduction

As of December 2021, the SARS-CoV-2 pandemic has led to over 300 million confirmed cases and more than 5 million confirmed deaths worldwide [1]. Mathematical modeling has been instrumental in understanding transmission dynamics, as well as in evaluating the impact of both pharmaceutical and non-pharmaceutical interventions [2–9]. A large number of different models were developed to account for the multitude of factors that were found to be important for the spread and control of SARS-CoV-2, including age [10, 11], seasonality [12, 13], and superspreading [14–20].

Through the analysis of contact tracing data and the reconstruction of transmission clusters, superspreading events have been shown to be a driving factor in the spread of several pathogens, among which are SARS-CoV-1, MERS, and, more recently, SARS-CoV-2 [21–27]. This means that the number of secondary cases caused by a single infectious individual is subject to substantial inter-individual variation. In other words, a small number of infected persons generates the majority of new infections, while most infected individuals cause only very few to no secondary cases.

In 2005, Lloyd-Smith et al. [28] proposed a framework to study superspreading dynamics in which the expected number of secondary cases caused by an infected individual, i.e., the individual reproduction number, is represented by a random variable, following a distribution —Lloyd-Smith et al. use a Negative Binomial distribution—on the positive real axis [28, 29].

As such, superspreading events can be characterized as occurrences in the right-hand tail of the aforementioned distribution. When such a superspreading event occurs, different factors, such as for example heterogeneity in infectiousness or heterogeneity in contact behavior, are at play [30]. Increased infectiousness—defined here as the likelihood that a social contact between an infectious and a susceptible individual leads to transmission—either due to behavioral, physical or biological reasons, may play a role. For example, more virus particles are shed when talking loudly or singing [31]. Furthermore, given the importance of aerosol transmission for the spread of SARS-CoV-2, differences in immune response among individuals may explain why some persons experience a more virulent infection than others, thereby excreting more virus particles and producing more secondary cases [32–34]. Additionally, timing is also important, i.e., individuals infected with SARS-CoV-2 are most infectious during a short interval [35], a period which is not necessarily accompanied by COVID-19 symptoms and associated behavioural change [36].

Some individuals have a higher number of contacts, and thus more opportunity to infect others, which is especially important during the short interval in which they are most infectious [37]. Some persons might not have a higher number of total contacts, but meet more susceptible individuals than others—such as for example in nursing homes [38]. Finally, the environment also plays an important role in the genesis of superspreading events: the risk of transmission is much higher in enclosed spaces than it is outside, and ventilation works well to limit transmission indoors [38, 39].

In the research presented here, we focus on inter-individual heterogeneity in infectiousness and heterogeneity in number of contacts. However, the other sources of heterogeneity mentioned above—intra-individual temporal variation in infectiousness, heterogeneity in susceptibility, and the environments in which contacts take place—remain an important topic for future research regarding superspreading.

Some transmission models for SARS-CoV-2 [14–20, 40], as well as for other pathogens like SARS-CoV-1 [41–43] and MERS [44], have taken superspreading into account. These approaches include compartmental models [15, 16], that divide the population in different and exclusive subpopulations based on realistic disease states, and branching process models [14, 28], as well as network and individual-based models [17–20, 40–44], in which each individual in the population is represented as a separate entity. Some models include the superspreading potential of individuals as a general factor [14, 17, 28], while others implement superspreading events as the result of heterogeneity in infectiousness [16], contact behavior [18, 19, 43], or a combination of both [15, 20, 41, 42, 44].

However, while some of these models do demonstrate that both factors contribute to the occurrence of superspreading events [20], it is still poorly understood how exactly these different forms of superspreading impact the spread of disease and the effectiveness of control measures. To further investigate these questions in detail, we tested the effect of different forms of superspreading on the spread of SARS-CoV-2 in the Belgian population. To this end, we implemented both infectiousness-related and contact-related heterogeneity into STRIDE, an individual-based model for the transmission of SARS-CoV-2 [9, 45].

In an individual-based model, each individual is represented by a separate entity with a unique set of characteristics, such as age, health status, and behavioral traits. As such, individual-based models are particularly suited to model superspreading, as they allow for a direct integration of different sources of heterogeneity at the individual level.

We investigated the effects of superspreading caused by variation in infectiousness versus heterogeneity in contact behaviour, in the absence of intervention measures. Additionally, we looked at the impact of different modes of superspreading on the effectiveness of social

distancing—a non-pharmaceutical intervention which continues to play an important role in the control of SARS-CoV-2.

## Materials and methods

### Implementation

We used STRIDE [45], an individual-based, stochastic model, which was recently adapted to encompass the disease-specific features related to the transmission of SARS-CoV-2 in the Belgian population [9]. In this model, individuals are assigned to social contact pools representing their household, a workplace or school (depending on the individual's age and employment status), and more general communities representing an individual's leisure contacts, contacts on public transportation and contacts in other locations. These latter communities consist of on average 500 persons and differ for each person between week- and weekend days.

Simulations move forward in discrete time-steps of one day. Each simulation day, two processes take place. First, the health state of each individual is updated. Individuals can be in one of the following states: Susceptible, Exposed, Infectious, Infectious and Symptomatic, Symptomatic, and Recovered. We assume that individuals obtain immunity after recovery from the disease, and maintain this immunity for the duration of the simulation. For more information on our implementation of the natural history of SARS-CoV-2, see S1 Text.

Secondly, individuals can access the pools of which they are a member, depending on the day of the week and their health status—e.g., children do not visit their school pool during the weekend, and symptomatic individuals primarily remain within their household pool. The number of contacts an individual makes each day in each of the above locations is based on their age and the type of location (household, school, workplace, or community). More specifically, daily contact rates are based on a social contact survey conducted in Belgium in 2010–2011 [46]. When a contact occurs between an infectious individual and a susceptible individual, a 'transmission probability' determines whether a transmission event occurs. This transmission probability depends solely on the infectious individual in our model, and not on the location in which the contact takes place.

We adapted the model to represent both infectiousness-related and contact-related heterogeneity. Both are implemented on the individual level: an individual will have a tendency to be more or less infectious, or have more or less daily contacts over the course of the entire simulation. Infectiousness-related heterogeneity was implemented as follows. When a person becomes infected, an 'individual transmission probability' is assigned to this person. This probability determines whether an actual transmission event takes place whenever a contact occurs between the infected individual and a susceptible individual. We assume that this probability remains the same for the entire duration of an individual's infectious period—one of the limitations we elaborate on in the discussion.

Following the work of Lloyd-Smith et al. [28], we chose to represent inter-individual variation by using a Gamma distribution. As such, each individual transmission probability is drawn from a (right-)truncated Gamma distribution on the interval (0, 1]. The truncated distribution is characterized by a shape parameter $\alpha_i$, which determines the level of overdispersion of the distribution, and a mean—hereafter referred to as the mean transmission probability. A lower value of $\alpha_i$ entails more variation in the individual transmission probability, and thus a higher level of superspreading, as shown in Fig 1A.

To account for contact-related heterogeneity, we multiply an individual's contact rate in community and workplace pools by a factor, which is unique for each individual. This factor is drawn from a Gamma distribution with shape parameter $\alpha_c$ and mean 1, upon creation of the

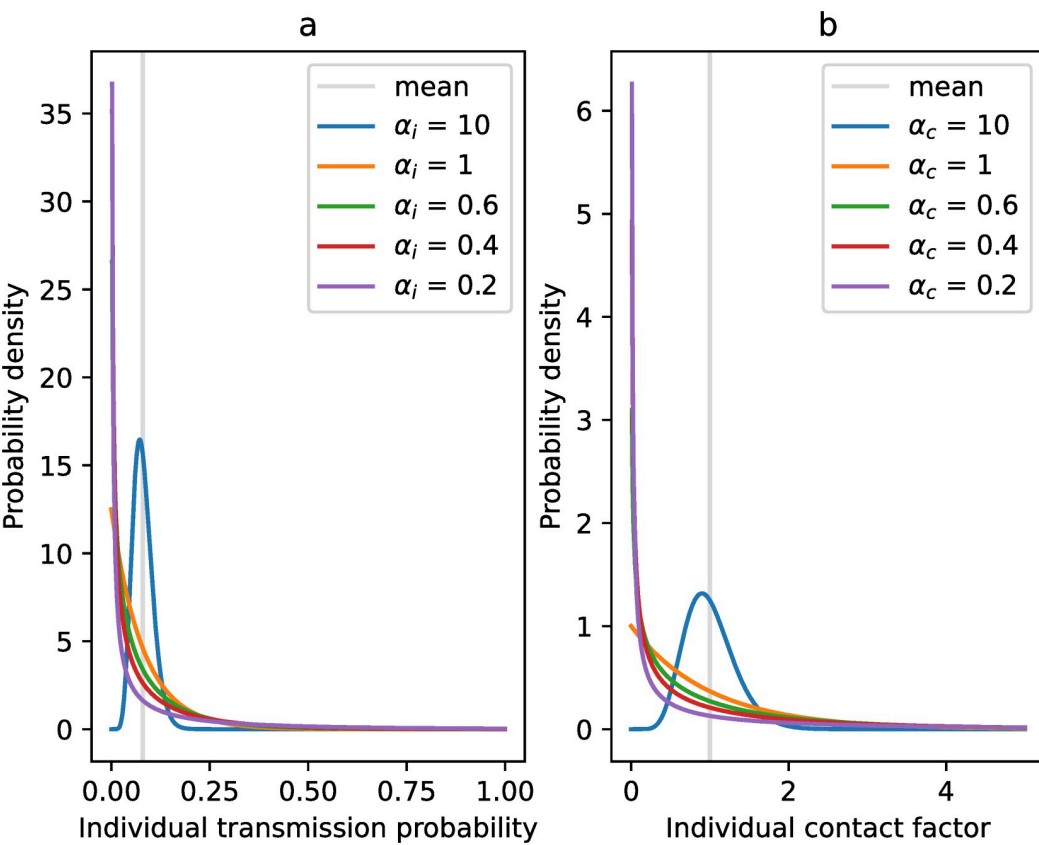

**Fig 1. Probability density functions of the Gamma distributions considered for the individual transmission probability (panel a) and the individual contact factor (panel b).**

individual. Again, a lower value of $\alpha_c$ implies more variation in contact rates and thus a higher level of superspreading, which is shown in Fig 1B.

The code of STRIDE is open-source and available at https://github.com/elisekaa/stride. All code and data used for the research in this paper has been assigned a DOI using Zenodo: 10.5281/zenodo.6669350.

## Simulations

For an exhaustive overview of the population and parameters used in the simulations described below, we refer to S1 Text and earlier work conducted on SARS-CoV-2 transmission using the STRIDE simulator [9]. As there is no standard procedure to estimate the number of realisations necessary for the type of simulations we conduct, we chose to run 200 stochastic simulations for each scenario described below, which led to stable results.

**Verification.**　We verified our implementation of superspreading in STRIDE in a number of ways. First, we confirmed that the mean $R_0$ remained stable over the different scenarios regarding infectiousness-related and contacts-related heterogeneity, to ensure that the differences we observed were indeed due to variations in infectiousness- and contact-related heterogeneity. We found that the mean $R_0$ was indeed consistent between the different scenarios, although the variance did differ—which we expected. This is shown in Fig A in S2 Text.

We also checked that the baseline case—in which both the individual transmission probability and the individual contact factor are constant—corresponds to a scenario in which either

the individual transmission probability or the individual contact probability follows a Gamma distribution with respectively $\alpha_i = 10$ and $\alpha_c = 10$. This should be the case, since, as the shape parameter $\alpha$ approaches infinity, the dispersion of the distribution decreases, and the distribution becomes more centered around the mean.

Furthermore, we calculated $P_{80}$: the minimal proportion of infected individuals that is responsible for 80% of cases—a measure commonly used in the superspreading literature [22, 28, 29]. We calculated $P_{80}$ for each simulation by ordering individuals that were infectious during the simulation according to the number of secondary cases they caused, in decreasing order. Next, we compute the proportion of individuals that are responsible for 80% of the total number of infections caused over the entire simulation run. We used this measure to verify that as we decreased either $\alpha_i$ or $\alpha_c$, this resulted in more heterogeneity in transmission—leading in turn to a lower $P_{80}$. We observed that this was the case, but the effect was stronger with the same values for $\alpha_i$ compared to $\alpha_c$. This is shown in Fig B in S2 Text.

Finally, we constructed a theoretical description for the transmission process in STRIDE, which allowed us to calculate the expected number of secondary cases per infected individual, as well as the variance of this quantity. We then compared the results of this theoretical description with simulation results for different forms and levels of superspreading. We observed that the results of our analytical calculations were largely in agreement with the results obtained through simulations, which is shown in Fig C–N in S2 Text.

For more details on the way in which these checks were conducted, and their results, see S2 Text.

**Superspreading effects in the absence of interventions.** First, we investigated the effect of different modes of superspreading on the unmitigated spread of SARS-CoV-2. To do this, we introduced one infected individual in a completely susceptible population at the beginning of a simulation and tracked transmission events over a period of 200 days. Typically, after this period, no more new infections were recorded for all considered scenarios, so we assumed we had observed most, if not all, of the epidemic curve. This is apparent from the plots depicting the evolution of the number of new cases and the cumulative number of cases per day, as shown in Fig B–E in S1 File.

To compare the effect of infectiousness-related heterogeneity to the effect of contact-related heterogeneity, we varied the distribution of the individual transmission probability and the individual contact factor as described in Table 1. The values for $\alpha_i$ and $\alpha_c$ were chosen as

**Table 1. Overview of scenarios.**

| Scenario | Individual transmission probability | Individual contact factor |
|---|---|---|
| Baseline | 0.08 | 1.00 |
| A | Trunc. Gamma ($\alpha_i = 10.0$, mean = 0.08) | 1.00 |
| B | Trunc. Gamma ($\alpha_i = 1.0$, mean = 0.08) | 1.00 |
| C | Trunc. Gamma ($\alpha_i = 0.6$, mean = 0.08) | 1.00 |
| D | Trunc. Gamma ($\alpha_i = 0.4$, mean = 0.08) | 1.00 |
| E | Trunc. Gamma ($\alpha_i = 0.2$, mean = 0.08) | 1.00 |
| F | 0.08 | Gamma ($\alpha_c = 10.0$, mean = 1.0) |
| G | 0.08 | Gamma ($\alpha_c = 1.0$, mean = 1.0) |
| H | 0.08 | Gamma ($\alpha_c = 0.6$, mean = 1.0) |
| I | 0.08 | Gamma ($\alpha_c = 0.4$, mean = 1.0) |
| J | 0.08 | Gamma ($\alpha_c = 0.2$, mean = 1.0) |

Overview of scenarios based on the distribution considered for the individual transmission probability and the individual contact factor.

follows: the value 10 represents a distribution in line with the baseline scenario (i.e., a distribution which approaches a degenerate one), with low levels of heterogeneity for either the individual transmission probability or the individual contact factor, a value of 1 was chosen as an intermediate scenario, and values 0.6, 0.4 and 0.2 were chosen to represent a high level of heterogeneity, in line with what we would expect to see in the transmission of SARS-CoV-2 [29]. The mean of the truncated Gamma distribution considered for the individual transmission probability (i.e., 0.08) was chosen so that it corresponds to an $R_0$ value of about 3 in the baseline scenario. This is in line with initial estimates for $R_0$ for SARS-CoV-2 [3, 9, 47, 48].

We then compared epidemiological metrics between the different scenarios. First, we looked at the probability of extinction, calculated as the fraction of simulation runs that produce no more new cases after only a few (or no) initial secondary infections [49]. We define what is regarded as extinction by looking at the final outbreak sizes after 200 simulation days, resulting from all simulations over all scenarios. After ordering these in descending order, a sharp drop-off can be observed, separating runs with persistent outbreaks from runs in which extinction occurs. As we found that outbreaks starting from one initial case either remain below 15 cases, or grow much larger, we set the threshold below which we will consider an outbreak to have gone extinct at 20 cases. In Fig A in S1 File, a histogram is shown depicting the frequency of final outbreak sizes per scenario.

We also compared the attack rate of outbreaks, and looked at the peak size and the timing of the peak. Furthermore, we investigated the evolution of the daily effective reproduction number $R_t$. We approximated $R_t$ by calculating the mean number of secondary cases caused by individuals that contracted infection on day $t$. However, since at the end of a simulation only a few infected individuals remain, thereby making inference about $R_t$ prone to substantial oscillations in daily estimates, we applied a LOWESS smoothing approach for the time-varying $R_t$-values [50].

To further gauge the effect of different forms of superspreading on epidemic spread, we estimated the herd immunity threshold, the day this threshold is reached, and the day on which the last infection is recorded. In the absence of a standard procedure to define the herd immunity threshold in an individual-based model, we estimated this measure as follows. We looked at the proportion of individuals that were no longer susceptible (i.e., recovered or currently infected) on the last day for which the smoothed $R_t \geq 1$.

Finally, we kept track of the type of location (household, school, workplace or community) in which infections occurred.

**Superspreading effects in the presence of social distancing.** We considered a scenario to investigate the impact of different forms of superspreading on the effectiveness of a period with strong social distancing followed by a period of mild relaxations.

The scenario was implemented as follows: after the introduction of a single infected individual in the population we simulated 30 days without interventions, after which a 'lockdown' period began, in which schools were closed (primary, secondary, and tertiary education), contacts at the workplace were reduced by 94.51% and contacts in the communities were reduced by 88.74%. These contact reductions were inferred based on social contact data collected for Belgium from April to mid-May 2020 in the CoMix study [51].

After 60 days of lockdown (i.e. on day 90 of the simulation), a partial release of the lockdown followed. Schools were re-opened and contacts in the workplace increased to 25.09% of pre-pandemic levels, while contacts in the community increased to 28.55% of pre-pandemic levels. Again, these contact reductions were estimated based on data collected in the CoMix study, from mid-May to August 2020. More information on how these and the above estimates were obtained can be found in S1 Text.

We tested the same distributions for the individual contact probability and for the individual contact factor as listed in Table 1. We ran each simulation for 600 days (30 days pre-lockdown, 60 days lockdown and 510 days partial release), after which we typically did not observe any new infections. This can be seen in the plots in Fig N–Q in S1 File, showing the number of new cases and the number of cumulative cases per day.

We compared the number of cases before, during, and after the lockdown between the different scenarios, as well as the attack rate over the entire simulation. We also looked at the evolution of the number of new cases, the cumulative number of cases and the effective $R_t$ per day. Additionally, we calculated a 'resurgence probability', to represent the chance that a lockdown followed by a partial release would not be effective in stopping the epidemic in a particular scenario. This resurgence probability was calculated as follows. As the effectiveness of a lockdown can only be gauged when the epidemic is still ongoing, and we did not consider the importation of new cases, we only took into account those runs in which new cases were still being observed when the lockdown period started. As such, we excluded simulation runs in which 0 cases were recorded during the entire lockdown period. Then, to define resurgence, we looked at the distribution of the number of cases during the period of partial release.

For all scenarios, we observed that either at most 309 cases occurred during the period of partial release, or a much larger number. As such we set the 'resurgence threshold' at 500 cases. In Fig L in S1 File, a histogram is shown depicting the frequency of numbers of cases during the release period per scenario.

**Sensitivity analysis.**   We conducted a sensitivity analysis regarding the mean transmission probability and the number of infected cases introduced at the beginning of the simulation. More information on this can be found in S3 Text.

To get an idea of the combined effect of infectiousness-related and contact-related heterogeneity on disease spread, we also ran simulations for combinations of $\alpha_i$ and $\alpha_c$. We constructed a grid using Latin Hypercube Sampling, and ran 50 simulations for each combination of $\alpha_i$ and $\alpha_c$ in this grid. We describe this analysis in more detail in S4 Text.

## Results

### Superspreading effects in the absence of interventions

We investigated the spread of SARS-CoV-2 throughout the Belgian population, following the introduction of one infected individual. First, we looked at the probability of extinction. Using the extinction threshold of 20 cases, we calculated, for each scenario, the proportion of simulation runs that lead to extinction. We observed that as infectiousness-related heterogeneity increases, the extinction probability also increases (see Fig 2A). In the baseline scenario, where the transmission probability is the same for all individuals, extinction occurs in 12.5% of all simulation runs. As $\alpha_i$ decreases to 0.2, however, we see that as much as 72% of the simulation runs produce less than 20 cases.

A different trend can be observed when varying contact-related heterogeneity (see Fig 2B). Here, varying $\alpha_c$ has very little effect on the fraction of runs that lead to extinction.

We also looked at the attack rate of outbreaks. In the baseline scenario, around 92.5% of the population is infected 200 days after the introduction of an infected individual in the population, whenever extinction did not occur. When increasing infectiousness-related heterogeneity as well as when increasing contact-related heterogeneity, the final size of outbreaks lowers. However, this decrease is much more pronounced in the case of contact heterogeneity. This is shown in Fig 3.

When looking at the size and timing of the peak of the epidemic curve of outbreaks when no extinction occurs (see Fig 4), opposite trends can be observed for increasing heterogeneity

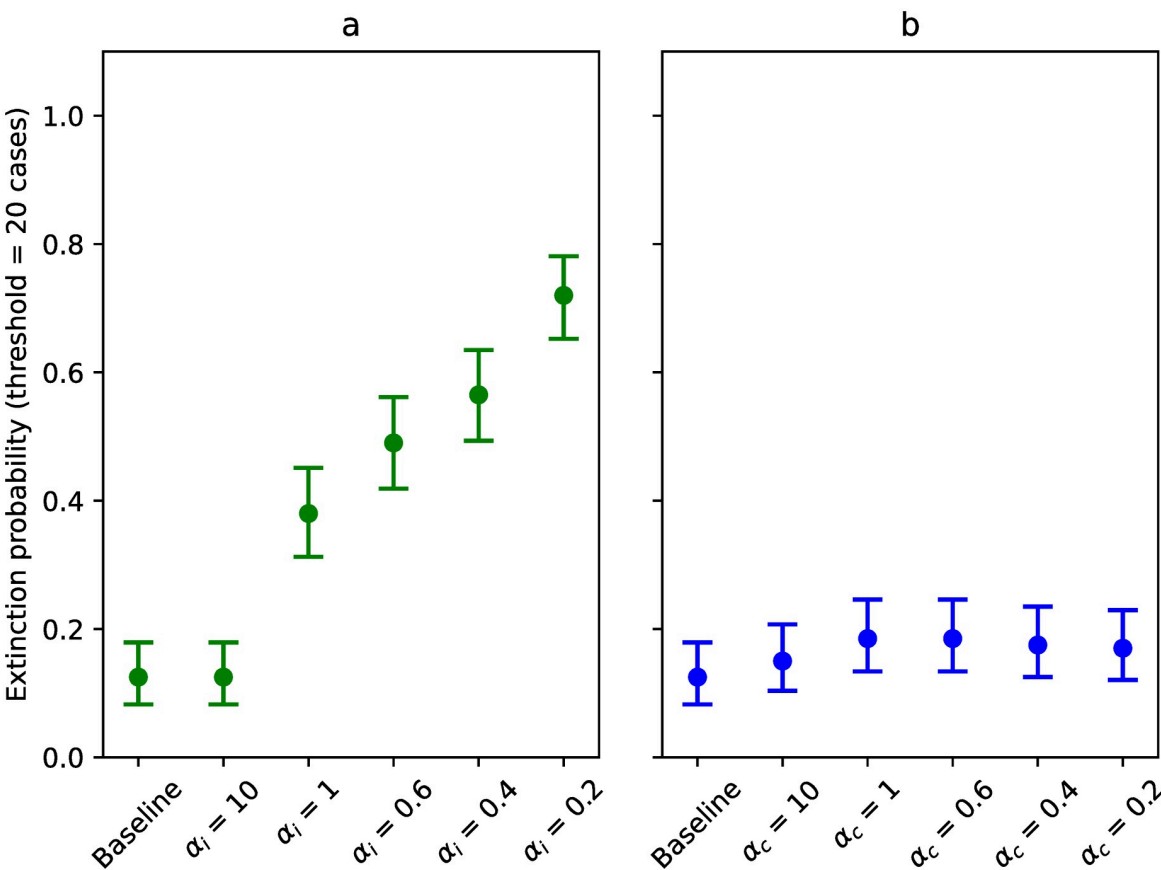

**Fig 2. Extinction probabilities for scenarios investigating infectiousness-related heterogeneity (in green, panel a) and contact-related heterogeneity (in blue, panel b), defined as the proportion of simulation runs ($n$ = 200) that produces less than 20 cases.** Error bars represent 95% (Clopper-Pearson) confidence intervals.

in infectiousness and contacts, respectively. When infectiousness-related heterogeneity increases, the mean size of the peak decreases, while no effect is seen on the time when the peak occurs. However, when contact-related heterogeneity increases, the mean peak size becomes higher and also occurs earlier. Furthermore, contact-related heterogeneity leads to more variability in peak size.

These trends can also be observed when looking at the evolution of the number of new cases, the number of cumulative cases and the smoothed daily $R_t$ values. The number of new cases per day, the cumulative number of cases per day, and the smoothed $R_t$ per day for the different scenarios is shown in Fig B–G in S1 File.

We estimated the impact of different modes of superspreading on the herd immunity threshold (see Fig 5). We observed that when infectiousness-related heterogeneity increases as $\alpha_i$ decreases from 10 to 0.2, the average herd immunity threshold decreases from 58.97% to 32.66%. This strong decline is in part due to the increase in the number of runs in which extinction occurs, but is still present when these runs are excluded, as can be seen in Fig 5C.

However, when contact-related heterogeneity increases, an opposite trend can be observed: the herd immunity threshold increases to 64.28% when $\alpha_c$ decreases to 0.2.

We also looked at the day on which the herd immunity threshold is reached (see Fig H in S1 File), and the day on which the last transmission event is observed (see Fig I in S1 File). We

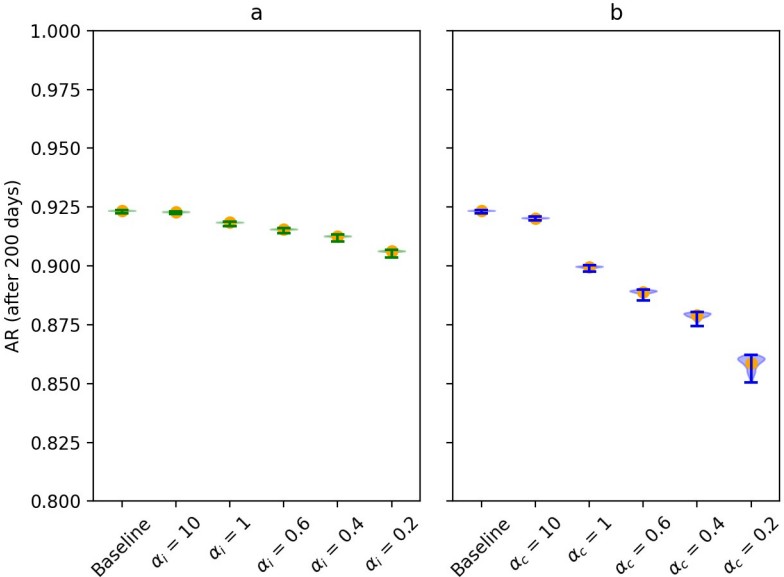

**Fig 3. Violin plots for the attack rate over 200 days for scenarios investigating the infectiousness-related heterogeneity (in green, panel a) and contact-related heterogeneity (in blue, panel b).** The orange dots represent the mean attack rate across the simulation runs without extinction, i.e., simulation runs in which extinction occurs ($< 20$ cases) were excluded.

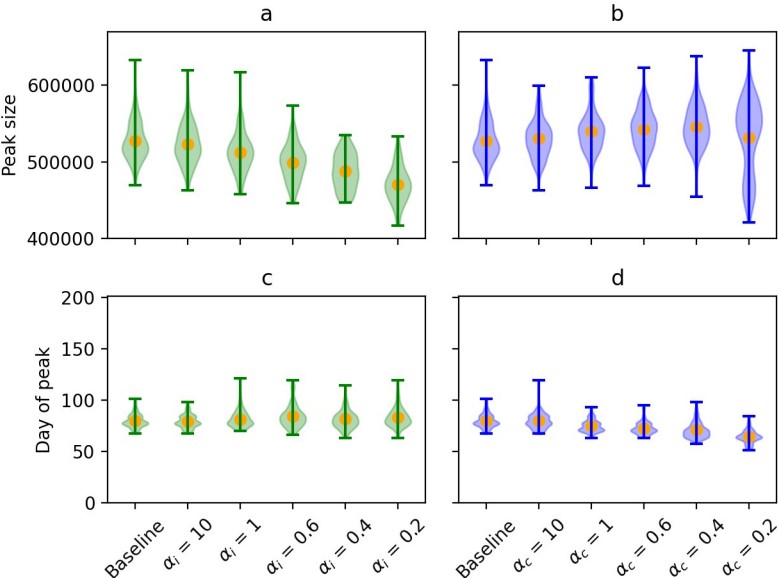

**Fig 4. Violin plots for the size and day of peak across the different simulation runs for scenarios investigating the infectiousness-related heterogeneity (in green, panels a and c, respectively) and contact-related heterogeneity (in blue, panels b and d, respectively).** The orange dots represent the mean of the simulated peak sizes and days at which the peak is reached. Simulation runs in which extinction occurs ($< 20$ cases) were excluded.

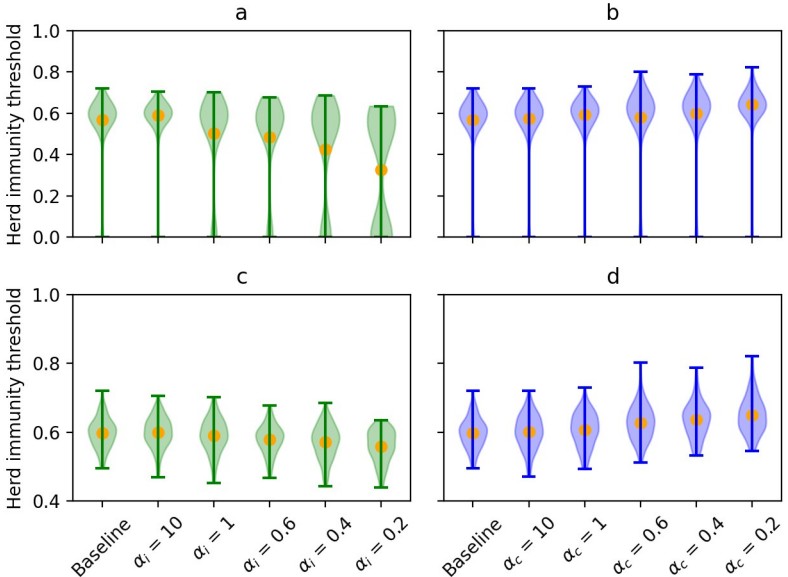

**Fig 5. Violin plots of the estimated herd immunity threshold values over the different simulation runs including or excluding runs with extinction for scenarios investigating the infectiousness-related heterogeneity (in green, panels a and c, respectively) and contact-related heterogeneity (in blue, panels b and d, respectively).** The orange dots represent the mean of the simulated values.

observed that, with higher infectiousness-related heterogeneity, the epidemic slows down, as the herd immunity threshold is reached later—even though it is lower—and the last infections are observed at a later time compared to the baseline scenario. Conversely, with higher contact-related heterogeneity, the herd immunity threshold is reached faster and infections stop occurring at an earlier point in the simulation.

Finally, we looked at the type of locations in which transmissions occurred (see Fig J–K in S1 File). We found that neither increasing heterogeneity in infectiousness nor increasing heterogeneity in contact behavior changed the locations where most infections took place, which for all scenarios were communities and households.

We also investigated the impact of different combinations of $\alpha_i$ and $\alpha_c$. We found that the effects we observed when testing different values for $\alpha_i$ and $\alpha_c$ separately were largely conserved when we tested combinations of $\alpha_i$ and $\alpha_c$, which is shown in Fig B in S4 Text. The extinction probability is still dominated by $\alpha_i$ (increasing for lower values of $\alpha_i$), while there was no discernible effect of $\alpha_c$. Although the attack rate decreased for both lower values of $\alpha_i$ and $\alpha_c$, the impact of $\alpha_c$ on the attack rate was much larger. The effects of $\alpha_i$ and $\alpha_c$ on the size and timing of the peak, the herd immunity threshold and the day on which the last transmission event was observed remained opposite. However, $\alpha_i$ had a bigger impact on the size of the peak and the herd immunity threshold, while $\alpha_c$ dominated the timing of the peak and the day of the last transmission event.

## Superspreading effects in the presence of social distancing

We investigated the effect of different forms of superspreading on the outcome of a social distancing scenario. We compared the number of cases before lockdown (day 0—day 30, see Fig 6A–6B), during lockdown (day 30—day 90, see Fig 6C–6D) and during the period of partial release (day 90—day 600, see Fig 6E–6F) between the different scenarios.

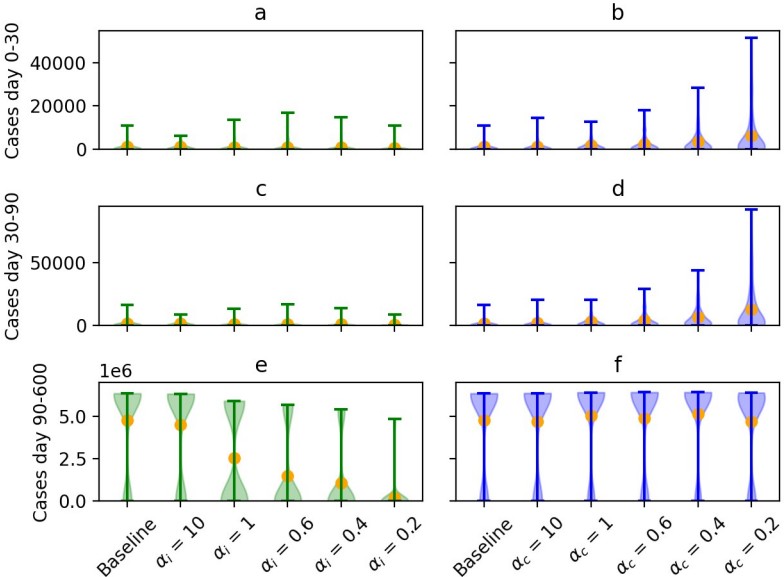

**Fig 6. Violin plots for the number of cases before lockdown (day 0–day 30), during lockdown (day 30–day 90) and during partial release phase (day 90–day 600) for the different scenarios investigating infectiousness-related heterogeneity (in green, panels a, c, and e, respectively) and contact-related heterogeneity (in blue, panels b, d, and f, respectively).** The orange dots represent the mean of the simulated values.

The number of cases observed before lockdown confirmed what we observed when examining unmitigated transmission: when contact-related heterogeneity is high, the start of outbreaks is more explosive. On the contrary, when infectiousness-related heterogeneity is high, many outbreaks stop after only a few cases. During lockdown, we also observe more cases when there is higher contact-related heterogeneity. However, as infectiousness-related heterogeneity increases, the average number of cases observed during lockdown slightly decreases. Finally, the number of cases during the partial release phase decreases sharply as infectiousness-related heterogeneity increases, while changes in contact-related heterogeneity seem to have little impact on the number of cases during this period. The same trends can be observed when looking at the attack rate over the entire simulation period (see Fig M in S1 File).

Furthermore, we looked at the evolution of the number of new cases per day, the cumulative number of cases and the smoothed daily $R_t$ values (see Fig N–S in S1 File). We observed that with higher infectiousness-related heterogeneity, outbreaks were slower to fade out during the period of partial release. When contact-related heterogeneity was high outbreaks were more explosive and took off again faster once social distancing measures were relaxed. However, what is remarkable here as well is that, as contact-related heterogeneity increases, distinct waves can be observed during the period of partial relaxations (see Fig M and Q in S1 File).

We calculated the resurgence probability for different forms and levels of superspreading (see Fig 7). As infectiousness-related heterogeneity increases, the resurgence probability decreases, meaning that it is less likely that the epidemic grows large in size again when relaxing measures after a period of lockdown. However, resurgence probabilities increase slightly as contact-related heterogeneity increases.

We assessed how the herd immunity threshold changes when social distancing is applied (see Fig 8). For the baseline case, the estimated herd immunity threshold lowered from 56.89% without interventions to 18.18% on average in the social distancing scenario. When increasing

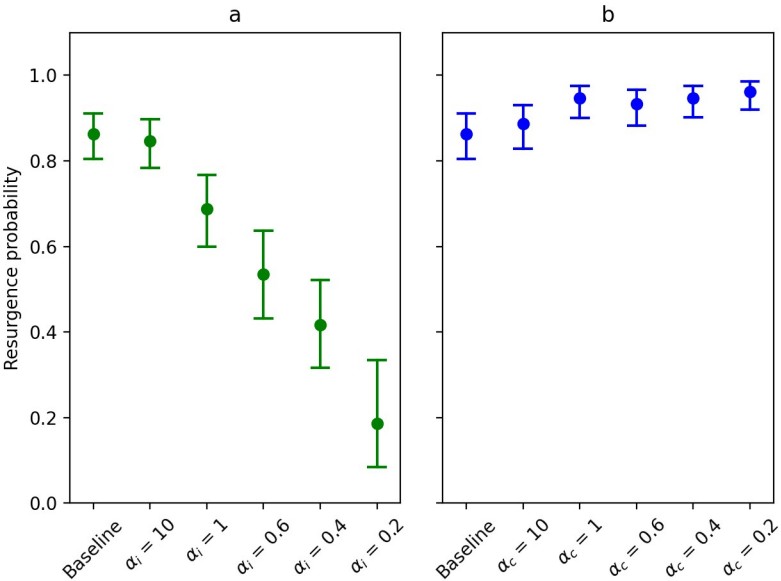

**Fig 7. The proportion of simulation runs, in which the number of cases during lockdown is greater than 0, that produces more than 500 cases during the partial release phase for scenarios investigating infectiousness-related heterogeneity (in green, panel a) and contact-related heterogeneity (in blue, panel b).** Error bars represent 95% (Clopper-Pearson) confidence intervals.

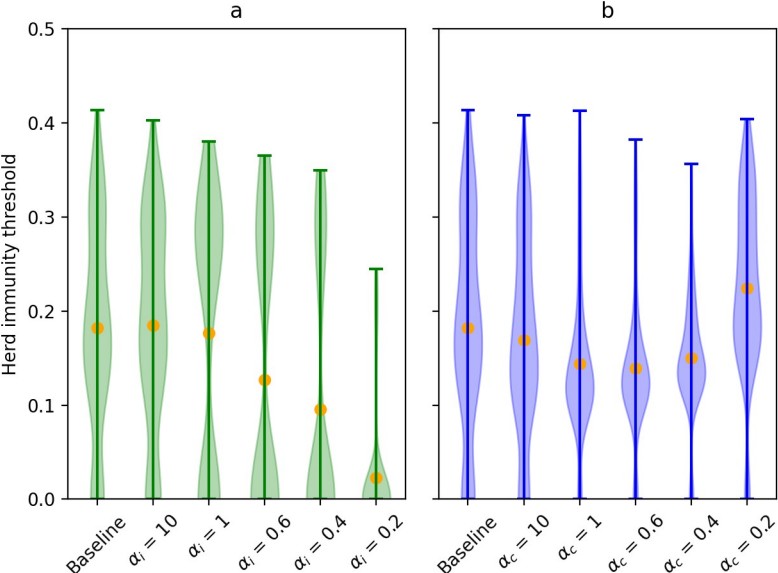

**Fig 8. Violin plots for the herd immunity threshold as estimated for the social distancing scenario, when investigating infectiousness-related heterogeneity (in green, panel a) and contact-related heterogeneity (in blue, panel b).** The orange dots represent the mean of the simulated values.

infectiousness-related heterogeneity, the herd immunity threshold decreased even further: to 2.28% when $\alpha_i$ = 0.2. However, this is not the case when contact-related heterogeneity is high. Even though the average herd immunity threshold initially lowers from 16.94% to 13.89% when $\alpha_c$ decreases from 10 to 0.6, it again increases to 22.43% when $\alpha_c$ is further decreased to 0.2.

Finally, we looked at the locations in which transmissions occurred (see Fig T–U in S1 File). We found that social distancing had little effect on where transmission happened: even though there were less transmissions overall, most transmissions still occurred in households and communities for all scenarios.

## Discussion

### Conclusion

We investigated different forms of superspreading in an individual-based model by considering both infectiousness-related heterogeneity and contact-related heterogeneity. We found that these two types of superspreading have very different effects, both on the unmitigated spread of SARS-CoV-2 as well as on the effectiveness of social distancing measures. In the absence of containment measures, we observed that with high infectiousness-related heterogeneity, the introduction of an infected individual in the population led to large outbreaks less frequently. Furthermore, peak sizes were smaller and occurred at a later time. The estimated herd immunity threshold also decreased. On the other hand, when contact-related heterogeneity is high, we observed that the introduction of an infected individual led to outbreaks slightly more often compared to the baseline scenario in which the mean transmission probability is 0.08 for all individuals and the individual contact factor is set to 1. Outbreaks that did not go extinct after only a few secondary cases were also more explosive, with higher peaks that occurred earlier. The herd immunity threshold also increased slightly.

The difference between the effects caused by these two forms of superspreading might be explained by the fact that when infectiousness-based heterogeneity is high, superspreaders have to infect other superspreaders to keep the epidemic going—leading to more extinctions and a drawn-out, but less explosive, outbreak whenever extinction does not occur. Furthermore, not all infected individuals with a very high individual transmission probability will have a lot of contacts and thus the opportunity to realize their 'transmission potential'. However, when a person has a large number of contacts, they not only have more opportunity to infect others once they are infected, but they also have a disproportionally high chance to be exposed to infection themselves [37], leading to faster, more explosive outbreaks.

We also observed that the total attack rate of outbreaks decreased much faster when contact-related heterogeneity increased than when infectiousness-related heterogeneity increased. This seems to be in contrast with the results described above, but can be explained due to a faster depletion of superspreaders (and susceptible individuals in general). This is enhanced by the fact that STRIDE uses a structured population, in which individuals are limited in the contacts they can make by the constraints of the contact pools (household, school, workplace, communities) they belong to. As such, when a large fraction of the persons in their contact pools has already been infected, an infected individual cannot cause a large number of secondary cases, no matter how infectious they are, or how many contacts they have [52]. Furthermore, as contact-related heterogeneity increases, more individuals will have no or very few contacts, making them 'unreachable', thereby decreasing the maximum number of cases that can be generated during an outbreak.

We should also note here that the same $\alpha_i$ and $\alpha_c$ parameters lead to different levels of over-dispersion in transmissions (as shown in Fig A–B in S2 Text)—which might explain some of the differences between both forms of superspreading.

Other models [16, 42, 44] that implement both heterogeneity in infectiousness and heterogeneity in contacts also conclude that superspreading driven by contact-related heterogeneity leads to more explosive outbreaks. These models (and others that only consider infectiousness-related heterogeneity), however, do not capture the slowdown of outbreaks due to infectiousness-related heterogeneity, since they do not implement a structured population, and hence every highly infectious person is able to realize their 'transmission potential'.

In this respect, some interesting parallels to research on network models can be drawn. A recent review considers the difference between 'out-degree' and 'in-degree' in partially directed networks [53]. Extinction probability is a function of out-degree, while epidemic size is a function of in-degree. This might explain why infectiousness-related heterogeneity has more impact on extinction while contact-related heterogeneity has more impact on attack rate: infectiousness-related heterogeneity mainly affects 'out-degree', while contact-related heterogeneity has more impact on 'in-degree'. An important difference however, is that the number of secondary cases an infected individual can cause in STRIDE is always limited by the contact pools that individual belongs to, no matter how large the total population is.

It is likely that in the current SARS-CoV-2 pandemic, a combination of both heterogeneity in infectiousness and heterogeneity in contacts—in addition to a multitude of other factors—has facilitated the occurrence of superspreading events [15, 20]. For example, for the original Wuhan SARS-CoV-2, $P_{80}$ was estimated to be about 0.1 [22]. Based on the results depicted in Fig A1 in S4 Text, we can then estimate $\alpha_i$ to be between 0.25 and 0.25, depending on the value for $\alpha_c$, which can be estimated from contact surveys. As circumstances change and new variants appear, the relative contribution of different forms of superspreading presumably also changes [54]. Calibrating our model of superspreading to the different waves and variants of SARS-CoV-2 would thus be a useful extension of our present work.

We also investigated the impact of different forms of superspreading on the effectiveness of social distancing. We found that, when superspreading is driven by heterogeneity in infectiousness, a period of strict social distancing, followed by a partial release is most effective and extinguishes almost all outbreaks. This is in line with the conclusion of other models that also assume superspreading events are driven by heterogeneity in infectiousness [18, 19].

However, when superspreading is driven by heterogeneity in contact behavior, we found that while social distancing measures might limit cases during a strict lockdown, the chance of resurgence of the epidemic following relaxations increases. Furthermore, with high infectiousness-related heterogeneity, few outbreaks grow large in size after a period of lockdown, but outbreaks that do not go extinct linger for a long time before completely disappearing.

## Limitations

Some limitations need to be taken into account when interpreting these results.

Each individual that is infected over the course of the simulation, eventually recovers and gains immunity for the remainder of the simulation. As such, neither deaths nor re-infections occur. We also did not model the waning of immunity, nor vaccination, making these results more representative of the initial wave of the SARS-CoV-2 pandemic. Additionally, we only introduce infected individuals at the beginning of the simulation, and do not take into account the continuous importation of infectious individuals into the population. Furthermore, we focus on the initial course of the infections, in which individuals can transmit the disease. We

did not model hospitalizations with case isolation, since we assume the transmission dynamics are mostly shaped by the pre-symptomatic and mild symptomatic stages and individuals exhibit adapted social contact patterns once they become symptomatic. To fully capture the effect of superspreading on the spread of SARS-CoV-2, these factors would, however, need to be taken into account.

To represent heterogeneity in infectiousness and contact behavior, we used a right-truncated and untruncated Gamma distribution respectively. However, other distributions could be used to obtain similar levels of heterogeneity, which we did not test in the current study [29].

Finally, we only modeled heterogeneity caused by differences between individuals: heterogeneity in infectiousness and heterogeneity in number of daily contacts. However, several other factors might contribute to the occurrence of superspreading events. These can be attributable to differences between individuals as well, such as heterogeneity in susceptibility [55] or infectiousness that varies over time [35], or they can be attributable to differences in the environment, such as the differing spreading potential between indoor and outdoor venues or superspreading facilitated by mass events [25, 56, 57]. Although these factors may be important in the occurrence of superspreading events, we did not model them in our current study. We did however model two different community pools (for weekdays and weekend days, respectively), where individuals could potentially contact a larger number (up to 500) of people, which could serve as a proxy for attending a large event. However, large events also have a temporal aspect that is not modeled through community contact pools. As such, explicitly modeling mass gatherings might lead to new insights regarding the effect of different types of superspreading on disease spread and on the effectiveness of interventions. Furthermore, our implementation of contact heterogeneity implicitly constitutes a proxy for heterogeneity in contact environment as well.

## Future work

Several extensions of this work could be useful in furthering our understanding of superspreading both within the context of the current COVID-19 pandemic and within a broader context.

Further research is needed to disentangle effects of both forms of superspreading. As such, it would be possible to estimate the level of both infectiousness-related and contact-related heterogeneity at play in the current SARS-CoV-2 pandemic. We plan to use data collected during the CoMix study [51] to estimate the level of contact-related heterogeneity during different phases of the SARS-CoV-2 pandemic. Subsequently, assuming that the two forms of heterogeneity investigated in this paper are most important for the occurrence of superspreading events, it is possible to estimate the level of infectiousness-related heterogeneity from the overall observed heterogeneity in transmissions. Further insight could also be gained by comparing the importance of these two types of superspreading for the spread of different pathogens, as contact-related heterogeneity presumably remains largely the same for a population under the spread of different pathogens.

We implemented heterogeneity in contact behavior by applying a distribution to the daily contact rate of individuals. There is however, in the context of social distancing, another possible type of contact heterogeneity, namely non-compliance to social distancing measures. It would be relevant to investigate how different forms of superspreading modulate the negative effect of this phenomenon on the effectiveness of social distancing measures.

Furthermore, while we investigated the impact of superspreading on the effectiveness of social distancing, it is conceivable that superspreading also has an impact on other

interventions, both pharmaceutical and non-pharmaceutical, such as vaccination, contact tracing and universal testing [14, 58].

Finally, we did not take into account person characteristics that might make an individual more or less likely to be a superspreader, due to the lack of data about this subject. Instead, the individual transmission probability and individual contact factor were drawn at random from a distribution. However, some characteristics, such as age, profession, or perception of the severity of COVID-19, might have an impact on how likely an individual is to transmit the disease [17]. Taking such characteristics into account when distributing transmission potential in the population would provide a more accurate model of superspreading.

## Supporting information

**S1 Text. Population and parameters.**
(PDF)

**S2 Text. Verification.**
(PDF)

**S3 Text. Sensitivity analysis.**
(PDF)

**S4 Text. Varying $\alpha_i$ and $\alpha_c$ together.**
(PDF)

**S1 File. Supplementary figures.**
(PDF)

## Acknowledgments

We used computational resources and services provided by the Flemish Supercomputer Centre (VSC), funded by the FWO and the Flemish Government.

## Author Contributions

**Conceptualization:** Elise J. Kuylen, Pieter J. K. Libin, Niel Hens.

**Data curation:** Lander Willem.

**Formal analysis:** Elise J. Kuylen, Andrea Torneri.

**Methodology:** Elise J. Kuylen, Andrea Torneri, Steven Abrams.

**Software:** Elise J. Kuylen, Lander Willem, Pieter J. K. Libin.

**Supervision:** Philippe Beutels, Jori Liesenborgs, Niel Hens.

**Validation:** Elise J. Kuylen, Steven Abrams, Pietro Coletti, Nicolas Franco, Frederik Verelst.

**Visualization:** Elise J. Kuylen.

**Writing – original draft:** Elise J. Kuylen.

**Writing – review & editing:** Elise J. Kuylen, Andrea Torneri, Lander Willem, Pieter J. K. Libin, Steven Abrams, Pietro Coletti, Nicolas Franco, Frederik Verelst, Philippe Beutels, Jori Liesenborgs, Niel Hens.

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
