## [Decision Letter · Decision Letter 0]

20 Apr 2022

Dear Mrs Kuylen,

Thank you very much for submitting your manuscript "Different forms of superspreading lead to different outcomes: heterogeneity in infectiousness and contact behavior relevant for the case of SARS-CoV-2" for consideration at PLOS Computational Biology. As with all papers reviewed by the journal, your manuscript was reviewed by members of the editorial board and by several independent reviewers. The reviewers appreciated the attention to an important topic. Based on the reviews, we are likely to accept this manuscript for publication, providing that you modify the manuscript according to the review recommendations.

Sincerely,

Miles P. Davenport, MB BS, D.Phil

Associate Editor

PLOS Computational Biology

Tom Britton

Deputy Editor

PLOS Computational Biology

[LINK]

Reviewer's Responses to Questions

**Comments to the Authors:**

Reviewer #1: Thank you for the opportunity to review this manuscript. In this study the authors applied an agent-based model for SARS-CoV-2 transmission, with two distinct mechanisms for superspreading implemented in the model: heterogeneity in the infectiousness of individuals (implemented by drawing the infectiousness of an infected person from a Gamma distribution); and heterogeneity in contact rates (implemented by drawing a multiplicative contact parameter for each infectious person again from a Gamma distribution). The authors compared the impact of allowing for each type of superspreading event in two types of transmission scenarios (one with unconstrained social mixing, and one with restrictions gradually being lifted). They found that each of these types of superspreading produces different effects when considering the probability of importation generating an epidemic; the size of the resulting peak; the overall attack rate; timing of the peak; and the herd immunity threshold. I thought that the authors reflection was interesting that the reason contact-based heterogeneity causes a larger peak and reduced chance of extinction is due to more initial opportunities for the epidemic to be seeded and agree that this seems to be the case.

I found this to be a well-written study, that was clearly explained, with well-presented results. I think it is a useful contribution to the literature on models that account for superspreading, and some of those points identified in the limitations would be interesting to see explored in future work. I have only minor comments and requests for clarification.

1. Abstract. “On the one hand, a higher level of infectiousness-related heterogeneity results in less outbreaks occurring following the introduction of one infected individual.”

Just to be very specific here – by fewer outbreaks, do you mean that the risk of an outbreak occurring following introduction of an infected individual into a susceptible population was lower? (From reading the methods I now see that this is the case but worth being clear in the Abstract).

2. How did you confirm that the overall level of transmission within each of the two superspreading scenarios is the same, when you were adjusting the infectiousness parameter versus the contact rate parameter (i.e. to ensure that you are comparing like-with-like)?

3. It took me a little while to identify in your paper what you assumed about acquisition and waning of immunity (as my immediate response to the results was to ponder the connection with immunity). I see in the first paragraph of your limitations section you do fully describe your assumptions but I think this should be made clearer earlier in your manuscript.

Reviewer #2: Kuylen et al. have used a preexisting COVID-19 simulation package to study 2 different "forms" of superspreading and determine the epidemiological consequences of the overdispersion in both factors in the population. My summary of the present results is: In a theoretical scenario with baseline contact related heterogeneity (CrH) but increased infectiousness-related heterogeneity (IrH), a simulation initiated by a single infected individual had a higher extinction probability, a delayed and decreased peak of infections, and a lower herd immunity threshold. In a parallel scenario with baseline IrH but increased CrH, peaks were earlier and higher, and herd immunity thresholds were higher. Resurgence after a lockdown increased with CrH but decreased with IrH.

The manuscript is very well written, and I found the results intuitive and the conclusions mostly justified by the results. I personally think the authors should do a bit more to take this work to a really high level -- it is already good but just misses out on some exciting opportunities and I don't think should be too much extra work.

My major comments are:

1) Most importantly, the conclusions could be strengthened and made much more generalizable by using a global sensitivity analysis. I think the current version teaches us how each mechanism works independently, but I think you really miss out on the opportunity to comment on what happens if both change in a scenario. If you can run some grid beyond Table 1 and try to see if these effects can cancel out (ie are not identifiable if like IrH goes up but CrH goes down), or if they can amplify in any surprising ways, this would really bring it to the next level. I know the means are varied in the supplement, but I think the heterogeneities and means should all be simultaneously varied and a typical global approach applied (see Sobol/Morris or more simple LHS protocols for example).

2) I thought there were some nice results in the supplement including agreement between simulations and analytical calculations. I thought maybe this is worth noting in the main body so that others could use your expressions

3)The definitions of infectiousness-related and contact-related still kind of miss the point you make about contact situation. Perhaps you should frame this as per-contact transmission probability heterogeneity, and allow this to contain the notion of more indoor gatherings?

4) I know you are trying to avoid "disentangling" these two effects, but I really think this misses an opportunity to tell a more applicable message. To me you have data in here that already help disentangle, for example: you could speculate on what you think happened between Alpha and Beta vs Omicron waves based on case data. Did the virus get more heterogeneous in terms of infection probability? or were people just tired of restritctions and more superspreading heterogeneity occurred? I think the case curves look pretty different, omicron was a lot sharper both up and down in a lot of places.

5) The figures were kind of unusual with the a) and b) legends above the actual legends. I think it could be a lot clearer (different colors or something) that left panels are always varying IrH and right are always varying CrH too

6) I didn't write much on text because in general it was clear and excellent, however, the abstract could be much stronger, see my comments below:

Abstract:

Superspreading events play an important role in the spread of SARS-CoV-2 and several other pathogens.

-Ok, but you don't really mention the other pathogens for the rest of the paper

Hence, while the basic reproduction number of the original Wuhan SARS-CoV-2 is estimated to be about 3 for Belgium, there is substantial inter-individual variation in the number of secondary cases each infected individual causes.

-Ok, this 3 seems oddly specific given that you don't talk about the magnitude of variation

Multiple factors contribute to the occurrence of superspreading events: heterogeneity in infectiousness and susceptibility, variations in contact behavior, and the environment in which transmission takes place.

-Why bring up 3 if you only study 2?

While superspreading has been included in several infectious disease transmission models, our understanding of the effect that these different forms of superspreading have on the spread of pathogens and the effectiveness of control measures remains limited.

-What do you mean? Need to be more specific about what you think is missing, why do we need to know more and what precisely is not known?

To disentangle the effects of infectiousness-related heterogeneity on the one hand and contact-related heterogeneity on the other, we implemented both forms of superspreading in an individual-based model describing the transmission and spread of SARS-CoV-2 in the Belgian population.

-"the Belgian population" is not defined

We considered its impact on viral spread as well as on the effectiveness of social distancing.

-I'm not sure you really studied superspreading vs social distancing, more like resurgence probabilities after social distancing restrictions are lifted. If you wanted to talk about effectiveness of social distancing, you might want to study how often you see case loads of X level despite social distancing.

We found that the effects of superspreading driven by heterogeneity in infectiousness are very different from the effects of superspreading driven by heterogeneity in contact behavior.

-"very" is not specific enough, since you don't present absolute numbers (which I think you should), you should at least present percent increases or something

On the one hand, a higher level of infectiousness-related heterogeneity results in less outbreaks occurring following the introduction of one infected individual. Outbreaks were also slower, with a lower peak which occurred at a later point in time, and a lower herd immunity threshold. Finally, the risk of resurgence of an outbreak following a period of lockdown decreased. On the other hand, when contact-related heterogeneity was high, this also led to smaller final sizes, but caused outbreaks to be more explosive in regard to other aspects (such as higher peaks which occurred earlier, and a higher herd immunity threshold). Finally, the risk of resurgence of an outbreak following a period of lockdown increased.

-fix less vs fewer

-is "outbreaks" really right? I'm not an epi specialist, but I think this word is kind of dangerous

-"also led to smaller final sizes" do you mean peaks? or is this prevalence? was prevalence lower after a simulatino with increased IrH

Determining the contribution of both source of heterogeneity is therefore important but left to be explored further.

-Like I said above, this just doesn't feel right, I think you can do a bit of this here instead of saving for later.

-Whatever you decide, this should not be the last sentence in your abstract, end on something strong about the utility of your study (which is really nice!) rather than a lame concession of its limitations

Reviewer #3: I’m a theorist and I have never used STRIDE, so I’m trying to understand these results in terms of known results in network epidemiology. As recently discussed in a review https://arxiv.org/abs/2005.11283, it’s important to distinguish forms of superspreading that affect the distribution of “out-degree” or “spread” - the number of secondary cases generated by a given infected individual - from those which affect to the “in-degree” or “risk” - the number of individuals from which a given individual can receive the infection. In some cases these are the same, since infections can spread in either direction along a social contact. But especially when different kinds of heterogeneity and interventions are taken into account, it’s important to think about the network of contacts as partially directed. For instance, an individual with high viral load would have a high out-degree, while an immunocompromised and highly susceptible person would have high in-degree. As another example, masking is often claimed to reduce out-degree more than in-degree (i.e. protect others from the masked individual more than the other way around).

In particular, in networks where short loops are rare, the extinction probability is a function only of the out-degree distribution, while the epidemic size (and therefore the attack rate if it is measured over the entire epidemic) is a function only of the in-degree distribution. As a result, models of superspreading where individuals generate differing numbers of secondary cases, but where these secondary infections fall uniformly on all individuals in a given compartment, don’t affect the epidemic size at all as long as R_0 is held constant. But they do affect the extinction probability, which increases as the variance in out-degree increases. Similarly, models where individuals have differing susceptibilities, but similar distributions of how many others they infect if they become infected, affect the epidemic size but not the extinction probability or epidemic threshold (again assuming R_0 is held constant).

So I’m trying to interpret Figures 2 and 3 in that context. Based on the analysis in https://arxiv.org/abs/2005.11283, it looks to me as if your infectiousness heterogeneity affects the “out-degree” or “spread”, while your contact heterogeneity mainly affects the “in-degree” or “risk”. This would explain why the extinction probability increases with infectiousness heterogeneity alpha_i but is essentially constant with respect to contact heterogeneity alpha_c. (In the text you claim a slight increase and then a downward trend, but the error bars in Fig. 2b are consistent with no dependence on alpha_c at all.)

Similarly, in Fig. 3 the attack rate depends strongly on alpha_c, as we would expect if alpha_c affects the variance of the in-degree, but much less strongly on alpha_i. Is it possible that if you waited longer, so that you observed the entire epidemic curve — making the attack rate equal to the epidemic size — that this dependence would disappear completely?

On the other hand, in the Conclusion you say “However, when a person has a large number of contacts, they not only have more opportunity to infect others once they are infected, but they also have a disproportionally high chance to be exposed to infection themselves.” So does your contact heterogeneity affect both risk and spread? I.e. does it increase the number of bidirectional contacts, along which the disease can spread in either direction? If so, this seems inconsistent with some of the experimental results.

As for the size and timing of the peak, I would argue that — if we condition on extinction not occurring — in the limit of large population size neither alpha_i nor alpha_c should make a difference, since once there are O(n) infected individuals the usual differential equation analysis should take over. In that case, the height and timing of the peak should depend only on R_0 and the matrix of mean contact rates between compartments, not on the distributions of contacts among individuals. Thus my guess is that the dependence in Fig. 4 is a finite-population effect.

That said, it is also true in branching process models that, if the variance in the degree distribution is large and we condition on the event that the process does not become extinct, then it has a higher average branching ratio at first (although this disappears after just a few generaitons if the underlying process is supercritical). This may explain your observation that the epidemic is more explosive at first — although I’m confused because I would expect this to depend more on alpha_i than alpha_c.

So to conclude: you have observed interesting distinctions between two types of superspreading in a popular epidemic simulation, and how they affect extinction probabilities, attack rates, and peaks in different ways. But since we know a lot about how epidemics depend on the structure of directed networks, it would be good to add a little more theoretical insight (which I think is available). That way the paper would be more convincingly about the effects of different mechanisms for superspreading, as opposed to just an observation about a particular simulation (i.e. STRIDE).

minor comments:

Using a right-truncated Gamma distribution for the individual transmission probability seems like a strange choice. Why not use, say, a Beta distribution, which also lets you choose both the mean and the variance while staying within the unit interval?

“as the overdispersion parameter approaches infinity, the distribution becomes more centered around the mean.” This phrase strikes me as strange, since as \\alpha increases the distribution has less dispersion, not more. Perhaps “shape parameter” is clearer.

**Have the authors made all data and (if applicable) computational code underlying the findings in their manuscript fully available?**

Reviewer #1: Yes

Reviewer #2: Yes

Reviewer #3: Yes

PLOS authors have the option to publish the peer review history of their article (what does this mean?). If published, this will include your full peer review and any attached files.

Reviewer #1: No

Reviewer #2: **Yes: **Daniel B Reeves

Reviewer #3: **Yes: **Cristopher Moore

Figure Files:

Data Requirements:

Reproducibility:

References:

---

## [Editor Report · Decision Letter 1]

29 Jun 2022

Dear Mrs Kuylen,

We are pleased to inform you that your manuscript 'Different forms of superspreading lead to different outcomes: heterogeneity in infectiousness and contact behavior relevant for the case of SARS-CoV-2' has been provisionally accepted for publication in PLOS Computational Biology.

Best regards,

Miles P. Davenport, MB BS, D.Phil

Associate Editor

PLOS Computational Biology

Tom Britton

Deputy Editor

PLOS Computational Biology

---

## [Editor Report · Acceptance letter]

17 Aug 2022

PCOMPBIOL-D-22-00314R1 

Different forms of superspreading lead to different outcomes: heterogeneity in infectiousness and contact behavior relevant for the case of SARS-CoV-2

Dear Dr Kuylen,

I am pleased to inform you that your manuscript has been formally accepted for publication in PLOS Computational Biology. Your manuscript is now with our production department and you will be notified of the publication date in due course.

With kind regards,

Andrea Szabo
